# Persistent Homology Captures the Generalization of Neural Networks Without A Validation Set

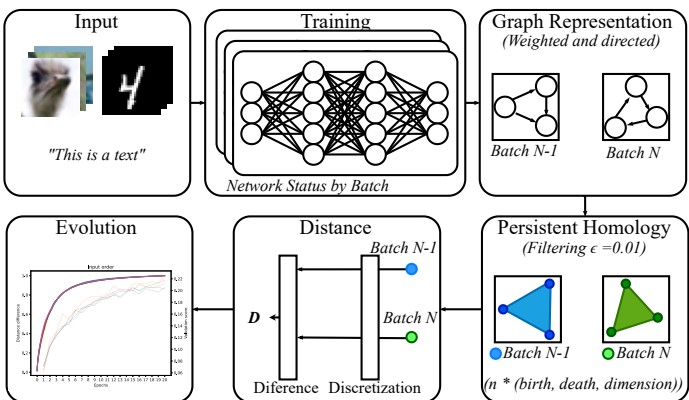

Figure 1: Our proposal.

## Abstract

The training of neural networks is usually monitored with a validation (holdout) set to estimate the generalization of the model. This is done instead of measuring intrinsic properties of the model to determine whether it is learning appropriately. In this work, we suggest studying the training of neural networks with Algebraic Topology, specifically Persistent Homology (PH). Using simplicial complex representations of neural networks, we study the PH diagram distance evolution on the neural network learning process with different architectures and several datasets. Results show that the PH diagram distance between consecutive neural network states correlates with the validation accuracy, implying that the generalization error of a neural network could be intrinsically estimated without any holdout set.

## 1 Introduction

Generalization is what makes a machine learning model useful; the uncertainty of its behaviour with unseen data is what makes it potentially dangerous. Thus, understanding the generalization error of a model can be considered one of the holy grails of the entire machine learning field.

Machine learning practitioners typically monitor some metrics of the model to estimate its generalization error and stop the training even before the numerical convergence to prevent the overfitting of the model. Usually, the error measure or the metric relevant to the task is computed for a holdout set, the validation set. Since these data have not been directly used for updating the parameters, it is assumed that the performance of the model on the validation set can be used as a proxy of the

generalization error, provided it is representative of the data that will be used in inference. One can, though, potentially overfit to this holdout set if is repeatedly used for guiding a hyperparameter search.

Instead of relying on an external set, though, the question of whether it could be possible to estimate the generalization error with some intrinsic property of the model is highly relevant, and it has been barely explored in the literature. On the other hand, Algebraic Topology has recently been gaining momentum as a mathematical tool for studying graphs, machine learning algorithms, and data.

In this work, we have the goal of, once having characterized neural networks as weighted, acyclic graphs, represented as Algebraic Topology objects (following previous works), computing distances between consecutive neural network states. More specifically, we can calculate the Persistent Homology (PH) diagram distances between a give state (i.e., when having a specific weights during the training process) and the next one (i.e., after having updated the weights in a training step) (see Figure 1. We observe that during the training procedure of neural networks we can measure this distance in each learning step, and show that there exists a high correlation with the corresponding validation accuracy of the model. We do so in a diverse set of deep learning benchmarks and model hyperparameters. This shines light on the question of whether the generalization error could be estimated from intrinsic properties of the model, and opens the path towards a better theoretical understanding of the dynamics of the training of neural networks.

In summary, our contributions are as follows:

- Based on principles of Algebraic Topology, we propose measuring the distances (Silhouette and Heat) between the PH persistence diagrams obtained from a given state of a neural network during the training procedure and the one in the immediately previous weights update.

- We empirically show that the evolution of these measures during training correlate with the accuracy in the validation set. We do so in diverse benchmarks (MNIST, CIFAR10, CIFAR100, Reuters text classification), and models (MLPs in MNIST and Reuters, MLPs and CNNs in CIFAR100 and CIFAR100).

- We thus provide empirical proof of the fact that valuable information related to the learning process of neural networks can be obtained from PH distances between persistence diagrams (*homological convergence*). In particular, we show that homological convergence is related to learning process and the generalization properties of neural networks.

- In practice, we provide a new tool for monitoring the training of neural networks, and open the path to estimating their generalization error without a validation set.

The remainder of this article is as follows. In Section 2 we describe the theoretical background of our proposal in terms of Algebraic Topology, while in Section 3 we go through the related work. Then, in Section 4 we formalize our method. Finally, in sections 6 and 7 we present and discuss our empirical results, respectively.

## 2    Background

In this section we introduce the mathematical foundations of this paper. A detailed mathematical description is included in the Supplementary Material.

A simplicial complex is a set composed of points, line segments, triangles, and their n-dimensional counterparts, named simplex ($K$). In particular, a simplicial complex must comply with two properties: 1. Every face of a simplex is also in the simplicial complex (of lower dimension). 2. The non-empty intersection of any two simplices contained on a simplicial complex is a face of both. 0,1,2,3-simplex and non simplex examples are shown in Figure 2.

We can associate to an undirected graph, $G = (V, E)$, a simplicial complex where all the vertices of G are the 0-simplex of the simplicial complex and the complete subgraphs with $i$ vertices, in $G$ corresponds to a $(i-1)$-simplex. This type of construction is usually called a complex clique on the graph G, and is denoted by $Cl(G)$. Figure 3 shows a graph clique complex Cl(G) example.

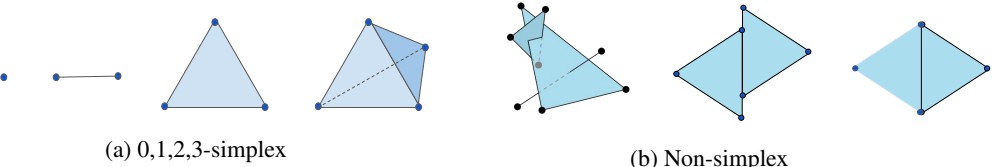

(a) 0,1,2,3-simplex        (b) Non-simplex

Figure 2: Simplex and non-simplex examples.

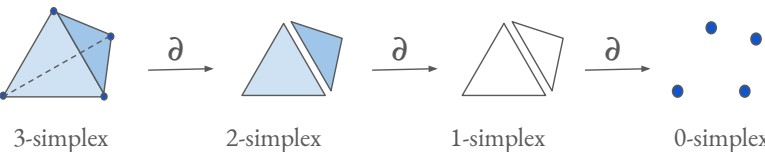

3-simplex     2-simplex     1-simplex     0-simplex

Figure 4: Boundary function sample.

The boundary function is defined as a map, from an $i$-simplex to an $(i-1)$-simplex, as the sum of its $(i-1)$-dimensional faces. A boundary function sample is shown in Figure 4.

In algebraic topology, a $k$-chain is a combination of $k$-simplices (sometimes symbolized as a linear combination of simplices that compose the chain). The boundary of a $k$-chain is a $(k-1)$-chain. It is the linear and signed combination of chain element boundary simplices. The space of $i$-chains is denoted by $C_i(K)$.

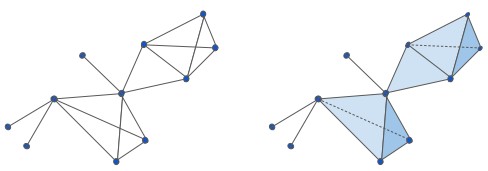

Figure 3: Graph clique complex Cl(G) example.

There are two special cases of chains that will be useful to define homology:

- Closed chain or $i$-cycle: $i$-chain with empty boundary. An $i$-chain $c$ is an $i$-cycle if and only if $\partial_i c = 0$, i.e. $c \in ker(\partial_i)$. This subspace of $C_i(K)$ is denoted as $\mathbb{Z}_i(K)$.

- Exact chain or $i$-boundary: An $i$-chain $c$ is an $i$-boundary if there exists an $(i+1)$-chain $d$ such that $c = \partial_{i+1}(d)$, i.e. $c \in im(\partial i+1)$. This subspace of $C_i(K)$, the set of all such i-boundaries forms, is denoted by $\mathbb{B}_i(K)$.

Now, if we think in the $i$-cycles that do not bound an $(i+1)$-simplicial complex, this is the definition $i$-th homology of the simplicial complex $K$. The precise definition is the quotient space of $\mathbb{B}_i(K)$ a subspace of $\mathbb{Z}_i(K)$ (see Supplementary Material). The number of non equivalent $i$-cycles (Figure 5) is the dimension of the homology group $H_i(K)$, also named Betti numbers.

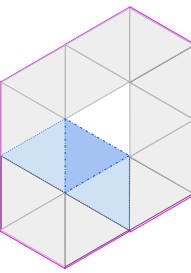

Figure 5: The two blue dashed cycles are homologically equivalent, the pink isn't.

We can create a nested family of simplicial complexes, $K_\varepsilon$, where at each step $t$, $K_{\varepsilon_t}$ is embedded in the simplicial complex $K_{\varepsilon_{t+1}}$. We call this set a simplicial complex filtration.

For each filtration simplicial complex, we can calculate the homology groups. Then, we can look at the birth, that is, when a homology class appears, and death, the time when the homology class disappears. The PH treats the birth and the death of these homological features in $K_\varepsilon$ for different $\varepsilon$ values. The lifespan of each homological feature can be represented as an interval $(birth, death)$, of the homological class. Given a filtration, this collection of intervals is named a Persistence Diagram (PD) [5].

It is possible to compare two PDs using specific distances (Wasserstein and Bottleneck). To efficiently perform this operation, due to the size of these diagrams, it is sometimes necessary to simplify them by means of a discretization process (such as Weighted Silhouette and Heat vectorizations).

## 3 Related Work

**Algebraic Topology and Machine Learning**  The use of Algebraic Topology in the fields of data science and machine learning has been gaining momentum in recent years (see Carlsson [5]). Specifically in the case of neural networks, some works have applied topology for improving the training procedure of the models [15, 8], or pruning the model afterwards [30]. Other works have focused on analyzing the capacity of neural networks [14, 26, 17] or the complexity of input data [17]. Furthermore, recent works have provided topological analysis of the decision boundaries of classifiers based on PH and Betti numbers [24, 22].

**Graph and topological representations of neural networks**  Gebhart et al. [12] suggest a method for computing the PH over the graphical activation neural networks, while Watanabe and Yamana [29] propose representing neural networks via simplicial complexes based on Taylor decomposition, from which one can compute the PH. Chowdhury et al. [7] show that directed homology can be used to represent MLPs. Anonymous [2] concurrently show neural networks, when represented as directed, acyclic graphs, can be associated to an Algebraic Topology object. By computing the PH diagram, one can effectively characterize neural networks, and even compute distances between two given neural networks, which can be used to measure their similarity. This is unlike other works [11, 13] approximating neural networks representations with regard to the input space.

**Estimating the generalization and studying the learning process**  We are, though, specifically interested in the use of PH for analyzing the learning process, especially with the goal of estimating generalization. In this regard, the literature is perhaps more limited. Jiang et al. [16] work on understanding what drives generalization in deep networks from a Bayesian of view. Neyshabur et al. [23] study the generalization gap prediction from the training data and network parameters using a margin distribution, which are the distances of training points to the decision boundary. In Li et al. [21], authors propose an alternative to cross-validation for model selection based on training once on the whole train set, without any data split, deriving a validation set with data augmentation.

Corneanu et al. [10] try to estimate the performance gap between training and testing using PH measures. They claim. However, one can observe some caveats. The first one is that their regression fitted to predict the test error has a considerably high error, making it not usable in practice. The second caveat is that for fitting the regression one needs at least part of the sequestered testing set.

In this work, motivated by the interest of having a better understanding of whether it would be possible to estimate the generalization of neural networks without a holdout set, we suggest using the topological characterization and distances concurrently proposed in Anonymous [2] but, crucially, measured between consecutive weight updates. We will show that the evolution of this distance is similar to the one of the validation accuracy. Unlike Li et al. [21], we do not use any data at all. Unlike [10], we do not build a statistical or machine learning model (linear regression) for predicting the testing error. Instead, we propose a new measure, and we empirically show that it highly correlates with the validation accuracy. Note that in this work we do not work with any input data and activations, but with the parameters of the neural network themselves.

## 4 Approach

**Representation**  For representing neural networks as graphs, we follow the approach proposed concurrently in Anonymous [2]. We associate to the neural network, at each learning state (defined by its weights), a weighted directed graph that is analyzed as an abstract simplicial complex. It is important to note that abstract simplicial complex are used in opposition to geometric simplicial complex.

For every training state, neural network connections are considered as directed and weighted edges between neurons, represented by graph nodes. Biases are considered as new edges that join to isolate vertices. In this representation, activation functions are lost. Bias information could also have been

ignored because, as we will see, it is not very informative in terms of homology, but we decided to preserve it.

Negative edge weights are represented with reverse edges with the same weight absolute value. We discard the use of weight absolute value as neural networks are not invariant under weight sign transformations. This representation is consistent with the fact that every neuron can be replaced by a neuron from which two edges with opposite weights emerge and converge again on another neuron with opposite weights. From an homological point of view, this would be represented as a closed cycle. Weights are normalized following the Equation 1. $\zeta$ is an smoothing parameter that we set to 1e-6. This smoothing parameter is necessary as we want to avoid normalized weights of edges to be 0 (in our representation 0 implies a lack of connection):

$$max(1 - \frac{|w|}{max(|max(W)|, |min(W)|)}, \zeta) \tag{1}$$

**Algebraic Topology object** For each weighted directed graph associated with the state of a neural network, we link a directed flag complex to it. The topological properties of this directed flag complex are studied using homology groups $H_n$. We calculate the homology groups up to degree 3 ($H_0$-$H_3$).

For each state, we use a family of simplicial complexes, $K_\varepsilon$, for a range of values of $\varepsilon \in \mathbb{R}$. The simplicial complex at step $\varepsilon_t$ is embedded in the complex at $\varepsilon_{t+1}$, for $\varepsilon_t \leq \varepsilon_{t+1}$, i.e. $K_\varepsilon \subseteq K_{\varepsilon_{t+1}}$. $\varepsilon$ is used as a filter that establish the minimum weight of the graph representation edges included on the simplicial complex. This collection of contained simplicial complex (associated to a directed weighted graph), called filtration, $K_{\varepsilon_{min}} \subseteq \ldots \subseteq K_{\varepsilon_t} \subseteq K_{\varepsilon_{t+1}} \subseteq \ldots \subseteq K_{\varepsilon_{max}}$, where $t \in [0, 1]$ and $\varepsilon_{min} = 0$, $\varepsilon_{max} = 1$ (remember that edge weights are normalized).

The sequence of homology groups is calculated by varying the $\varepsilon$ parameter to obtain the persistence homology diagram. In our case, persistent homology calculations are performed on $\mathbb{Z}_2$. In other words, once the corresponding filter has been applied to the weight of the edges, all connected edges are considered equally.

**Distances between persistence diagrams of consecutive states** In this paper, we are interested in comparing PDs between different simplicial complex associated to each training state of the neural network. There are two distances traditionally used to compare PDs, Bottleneck distance (the length of the longest edge) and Wasserstein distance (using the sum of all edges lengths, instead of the maximum). Their stability with respect to perturbations on PDs has been object of different studies [6, 9].

In order to make computations feasible and obviate noisy intervals, we filter the PDs by limiting the minimum PD interval size. We do so by setting a minimum threshold $\eta = 0.01$. Intervals with a lifespan under this value are not considered (spurious homological features). Additionally, for computing distances, we need to remove infinity values. As we are only interested in the deaths until the maximum weight value, we replace all the infinity values by 1.0.

In our case, our neural networks have millions of persistence intervals per Persistence Diagram, while Wasserstein distance calculations are computationally hard for large PDs. In order to make calculations computationally feasible, we will use a vectorized version of PDs, also called PD discretization. This vectorized version summaries have been proposed and used on recent literature [1, 3, 4, 19, 25]. For persistence diagram distance calculation, we use weighted Silhouette and Heat vectorizations, using the Giotto-TDA library [27].

# 5 Experiments

**Data** We validate our method in several heterogeneous (vision, natural language), well-known datasets, namely 1. MNIST [20], 2. CIFAR-10, 3. CIFAR-100 [18], and 4. the Reuters dataset [28] (multi-class and multi-label document classification dataset).

**Models** We experiment with two neural architectures,1. MLPs and 2. CNNs. In the latter case, we use the convolutional layers as a pre-trained model with frozen weights, and we learn an MLP on top of it. The reason we do so is that our method is based in a representation that, at least in the basic form, does not allow capturing information from convolutional layers. Thus, we need a single (exact

same weights) feature extractor, to abstract away distances related to the CNN layers and focus on the MLP.

**Conducted experiments** We define the *base* MLP architecture as {Input, Linear(512), Dropout(0.2), Linear(512), Dropout(0.2), Output}. In the case of CNNs, the pre-trained model is defined as 3 convolutional blocks with kernel size 3 (starting with 32 channels), interleaved with max pooling (its linear layers are thrown away after the pre-training). On top of the pre-trained CNN, we also define the same base MLP architecture. Then, for each dataset and model (MLP and CNN), we experiment with varying (while keeping the rest fixed to the base architecture)

1. Layer size (number of units per layer): 4, 16, 32, 128, 256.

2. Number of labels (the other classes are removed): 2, 4, 6, 8, 10.

3. Learning rate: 1e-e05, 0.0001, 0.001, 0.01, 0.1

4. Dropout: 0.0, 0.2, 0.4, 0.5, 0.8.

5. Input order: 5 random input orders. As a control experiment, for each analyzed problem we run the same configuration with 5 different input orders. If the measured distances are, indeed, related with the learning process of neural networks, these variations should not have any noticeable effect.

We run each configuration 5 times with different random seeds (and, thus, weight initializations[1]) to see if the results are consistent across runs. All models are trained with the RMSProp optimizer with a batch size of 256.

**Distances and validation accuracy computation** Note that homological distances are obtained at the end of each batch, while validation metrics are only computed on each epoch. The methodology we follow to analyze the learning process on each different problem can be summarized with the following steps:

1. In each training step (i.e., for each batch) we extract the weights from the MLP current state and use them to build an abstract simplicial complex from the associated weighted directed graph.

2. We calculate the homological persistence diagram of the simplicial complex.

3. We then calculate the distance between consecutive persistence diagrams (we will call this sequence *homological convergence*). We use two different distances, namely, Heat and Silhouette.

4. We compare the homological convergence with the evolution of the validation results on neural network learning process.

**Hardware** All experiments were executed in a machine with 2 NVIDIA V100 of 32GB, 2 Intel(R) Xeon(R) Platinum 8176 CPU @ 2.10GHz, and of 1.5TB RAM, for a total of around 7 days. We note that our method is considerably demanding in terms of both compute and memory.

The code and outputs are fully available in the Supplementary Material under MIT License.

# 6 Results

In this section, we highlight the main results, omitting the ones with Silhouette (since the obtained results were clearer with Heat). See the Supplementary Material for the full results (plots and correlations), including the ones with Silhouette distance.

We study the relation between the evolution of the PH diagram distances with the one of the validation score with the cumulative values of the distance between homologous persistence diagrams because this value seems much more stable. The information of the distance between the persistence diagrams has been normalized to visualize clearly the type of evolution of each curve on the same scale. Some of the non-normalized plots can be found in the Supplementary Material. Figure 6 shows the cumulative and non-cumulative homology the MNIST experiment with layer size.

---

[1]The pre-trained convolutional weights are always identical, though.

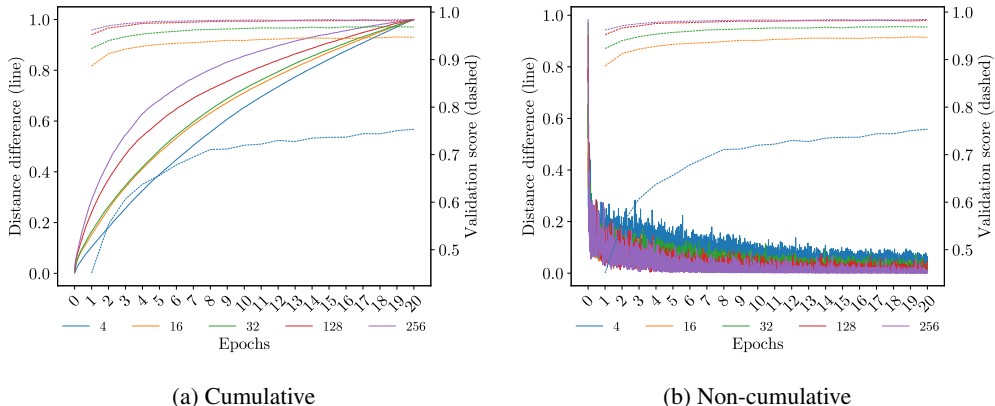

(a) Cumulative            (b) Non-cumulative

Figure 6: Heat distance and validation accuracy curves on the MNIST experiment with layer size. Normalized.

For each experiment (e.g., layer size in MNIST), we plot both the evolution of the PH diagram distance and the validation score (accuracy). The plotted values are the corresponding means of the 5 repetitions with different seeds. In addition, we compute the Pearson correlation for these values. Plots show on the x-axis each training step (for each batch) of the evolution in the training state of the neural network. On the y-axis, two scales are shown that apply to the distance curves between accumulated persistence diagrams (solid lines), scale on the right side, and the neural network validation (dotted lines), numerical scale on the left side. For each sub-experiment (for example, different values of layer size) a different color was used.

The general result is that the evolution of the homological convergence of the MLPs seems to be very similar to the one of the validation score. This is generally consistent across experiments (see the Supplementary Material). Table 1 shows the mean (and standard deviations) of the Pearson correlations for all datasets. All means are above 0.8, implying that there is strong correlation. Intuitively, this is also observed in the plots, although once the distances are normalized it is not as clear to visualize. Interestingly, we find that the very few exceptions in which the correlation is low corresponds to extreme values (very small number of neurons per layer, very high learning rate, very high dropout), in which the neural network doesn't end up learning properly.

In the case of CNNs, the correlations are lower (although still almost always above 0.8 in experiments such as the one of increasing the number of layers). Recall that in the case of CNN we froze a single convolutional feature extractor, since our method only supports MLPs. We believe these lower correlations can be explained because an important part of the learning process happened in the convolutional layers (in the pre-training), which we do not capture.

Another finding is that the method obtains consistent results across runs, meaning that it is capturing information related to important properties of the networks themselves instead of random artifacts.

When varying the studied hyperparameters, we observe that the curves for each configuration are indeed, different. Remarkably, in the control experiments, this is not the case; results show that the homological convergence during the learning of the same problem with the same model but with different input order is very similar. The alteration of the order of the input doesn't have any effect in the homological convergence. The results of two of these experiments are shown in Figure 7.

In addition, we observe that when the neural network learns the given problem, homological convergence occurs. For example, when the layer size is modified, the capacity of the neural network to learn the problem changes (Figure 6). When it can't learn the problem, because the network does not have sufficient capacity (the layer size is too small, 4 units), the homology does not seem to converge.

Regarding the learning rate, the results are coherent with the intuition that it is a fundamental parameter that controls how much to change the model in response to the estimated error during the learning process. A too small learning rate may result in a long training process that could be stalled, while a too large value may fall in a fast suboptimal solution or an unstable training process. Using homological convergence we find similar behaviour, as can be seen in Figure 9.

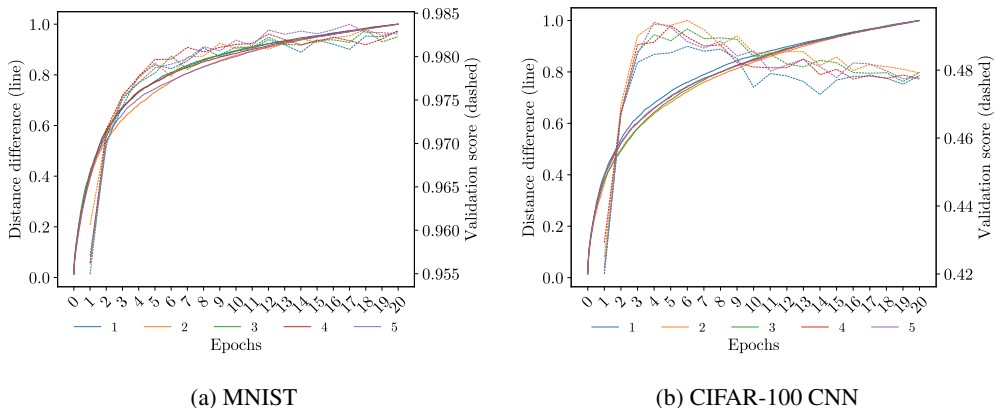

(a) MNIST             (b) CIFAR-100 CNN

Figure 7: Learning evolution on input order experiments (control experiments). Normalized.

| Dataset | Heat distance | | Silhouette distance | |
| --- | --- | --- | --- | --- |
| | Means mean | Deviations mean | Means mean | Deviations mean |
| MNIST | 0.8910 | 0.0424 | 0.8910 | 0.0424 |
| Reuters | 0.6220 | 0.0700 | 0.6220 | 0.0700 |
| CIFAR-10 MLP | 0.8233 | 0.0649 | 0.8233 | 0.0649 |
| CIFAR-10 CNN | 0.4241 | 0.1915 | 0.4241 | 0.1915 |
| CIFAR-100 MLP | 0.8420 | 0.0566 | 0.8420 | 0.0566 |
| CIFAR-100 CNN | 0.6130 | 0.0800 | 0.6130 | 0.0800 |

Table 1: Correlation of validation values with topological difference cumulative. Correlation is computed with 20 points.

Finally, we note that even if the two convergences (validation and homological convergence) are correlated, they are not the same process. This is especially visible in the case of the learning rate experiments. For instance, in Figure 9, homological convergence is reached before the stabilization of the validation accuracy. Presumably, they are not capturing the exact same information; specifically, we believe that the difference is due to the fact that the validation accuracy depends on the specifics of the data sampled in the validation subset, while the homological convergence is independent of the validation data.

## 7 Discussion

We posed the of question whether homological convergence (in terms of distances between PH diagrams in consecutive neural network states) is related to the learning process of neural networks. We have seen that, indeed, it is the case, with strong empirical results backing our claim.

This finding has a remarkable implication. If the homological convergence evolution mirrors the validation accuracy curve, one could ignore the validation set to monitor the training. This opens the path towards estimating the generalization of neural networks without the need of any holdout set. Researchers have wondered for a long time whether generalization could be predicted from intrinsic properties of the model or training data alone (i.e., without a holdout set), and in fact other works have claimed to do so. Although we do not provide any predictive model, we show that our proposed measures strongly correlate with the validation accuracy. In addition, we do so by not using any data at all; we just look at the neural network itself.

Our contribution aims pushing towards having a better understanding of the learning process of neural networks, not targeting any specific direct application. However, we note that it can be effectively used for monitoring the training of neural networks in terms of convergence expected generalization, as we have extensively shown in the experiments. Apart from the cases without access to a validation

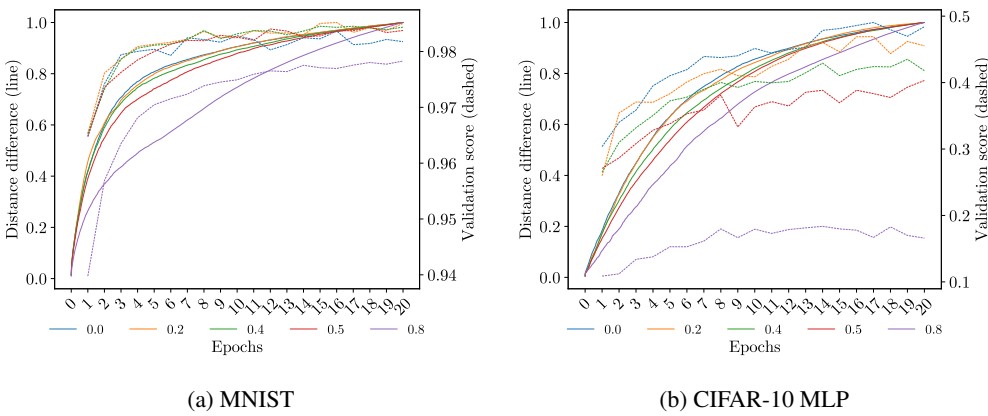

(a) MNIST

(b) CIFAR-10 MLP

Figure 8: Learning evolution when dropout parameter is changed. Normalized.

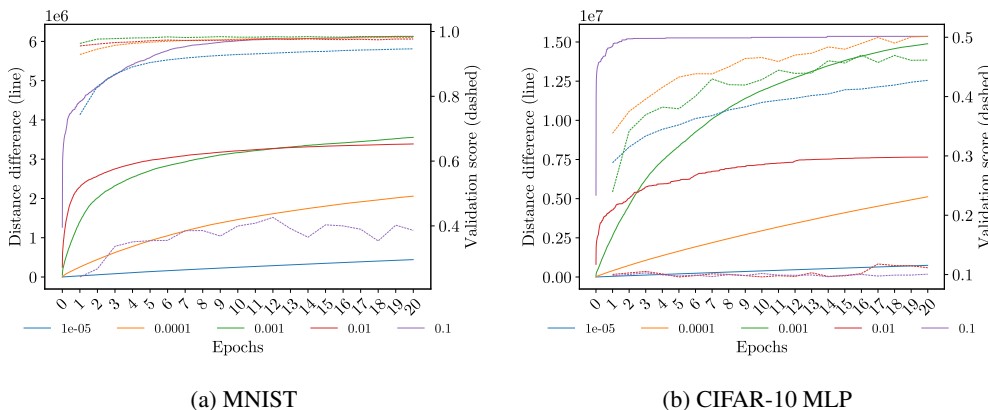

(a) MNIST

(b) CIFAR-10 MLP

Figure 9: Learning evolution when modifying the learning rate parameter. Not normalized.

set, this is relevant because depending on a validation set has the risk of overfitting to it. Having an intrinsic, well-principled measure should be more robust to random noise in a specific data sample.

The main limitation of our method is its computational scalability. As we said in Section 5, our method took more than 7 days of compute in a HPC machine, even if we restricted the experiments to small datasets and parameter count. However, we note that our approach computes the *exact* persistence diagram distances, that is, we do not simplify the graph representation of the neural networks (we keep every single neuron and connections) and we do not approximate any computation. This leaves room for finding efficient approximations, opening a new research line. In addition, this lack of scalability has prevented us from validating our method on bigger models and datasets.

Finally, we note that instead of computing correlations, serving as a basic quantitative study, it would be interesting to perform a time-series analysis to gain more insights on how the two curves vary together. Moreover, it would have been interesting to investigate how to build a predictive model of the validation accuracy from the PH distances, but it is was of the scope of this work.

## 8   Conclusions & Future Work

In this work, we have provided an empirical proof of the fact that homological convergence is related to the learning process and generalization properties of neural networks. Furthermore, we have shown that it can be used to monitor the training of a neural network (and potentially estimating its generalization) without a validation set. As future work, we suggest generalizing our representation to other neural architectures and scaling up the experiments to larger models and datasets, for which finding efficient approximations of our method will be crucial.

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
