# OpenReview forum: "Persistent Homology Captures the Generalization of Neural Networks Without A Validation Set"
_NeurIPS.cc/2021/Conference — NeurIPS 2021 Submitted_

### Official Review · Reviewer_Kand · 2021-07-05

**Rating:** 3
**Confidence:** 5

**Summary:**

This paper analyses the training of neural networks from a topological
perspective, presenting a pipeline that can measure (pseudo) distances
between the network's weights during training. Such information is then
employed to study the generalisation error of a neural network.

In contrast to existing methods for estimating this error, this paper
does *not* require a specific hold-out data set, as topological features
of the neural network are monitored during training. This frees up
additional data for fitting, which can be highly relevant in the sparse
data regime.


**Limitations And Societal Impact:**

Limitations of the method could be more prominently discussed. As in
previous work, the current method does not work well with convolutional
architectures; an additional subsection on limitations would therefore
be appreciated, in particular by readers that are not yet familiar with
TDA. This could be integrated into Section 7, which already contains
a discussion of the computational requirements.

Being mostly a theoretical analysis of generalisation performance, no
negative societal impact is bound to arise specifically from this work.

**Main Review:**

## Summary of the review

I enjoyed the ideas presented in this paper; the analysis of
generalisation performance is a highly relevant and timely.
Understanding generalisation without requiring additional data has the
potential to improve machine learning techniques to a significant
extent.

That being said, the current paper suffers from some issues, which
prevent my endorsement at this point:

1. Clarity: while background information on topological data analysis is
   provided (which I appreciate; in particular in light of the fact that
   TDA is still a rather novel occurrence at ML conferences), the method
   itself could be described in more detail. In particular, some aspects
   should be discussed in Section 4:

    - How does the calculation work in practice? The paper only provides
      a description of simplicial homology in the undirected setting but
      Section 4 appears to discuss directed graphs as well. I understand
      that the directed graph is turned into an undirected one, but this
      needs to be clarified.

    - Would it be possible to use a directed flag complex or a Dokwer
      complex here instead? Maybe this is more of an expert discussion
      topic, but if directionality matters, that could be a good option.

    - The use of additional representations for topological features
      needs to be clarified. At present, the main paper does not contain
      any description of the challenges in calculating distances or the
      need for additional representations. At the very least, a brief
      description of these topological representations is required, so
      that readers may better understand the remainder of the paper.
      I find it particularly problematic that no stability guarantees of
      these representations are discussed; they are not *just* drop-in
      replacements for the bottleneck or Wasserstein distances.

    - Overall, this section should be rewritten with a clear 'roadmap'
      in mind. What is the problem you want to tackle and how do you
      intend to tackle it in this paper?

2. Delineation to existing work: The paper by Rieck et al. [26] appears
   to already contain a large amount of the material proposed in this
   paper. A cursory reading shows that Rieck et al. even discuss
   generalisation performance in an early stopping setting (more about
   this later). It is therefore critical that the contributions of this
   paper are more clearly delineated from existing work. From my
   understanding of both papers, I would say that the current submission
   improves on the following aspects:

    - Choice of filtration for such neural networks

    - The use of other topological representations (whereas Rieck et al.
      only use a summary statistic of persistence diagrams).

  These differences (and potentially all others that I missed) should be
  briefly discussed; the advantage would be that the paper can refer to
  the previous publication as a justification of the method itself!

2. Experimental depth: the experimental section is lacking depth, in
   particular given the specific goals of this paper. The depicted plots
   and correlations are useful, but don't serve to highlight the
   benefits of the proposed method. I would therefore suggest a more
   thorough experimental setup such as the one shown in the supplement
   of Rieck et al. [26]: here, their proposed measure of network
   complexity is shown as additional early stopping criterion (without an
   hold-out validation data set as well).

   A post-hoc experiment of this sort would be extremely useful in
   demonstrating the utility of the method, and it would even enable the
   quantification of certain measures. The correlation is also useful in
   this context, but since the goal is to assess the generalisation
   error, an experiment in which the new measure is directly applied
   would be extremely worthwhile.

Please see below for detailed comments.

## Detailed comments

- The first contribution reads a little bit like a sentence fragment to
  me. I would suggest to use the terms 'silhouette distance' and 'heat
  kernel distance' instead of plain 'silhouette' and 'heat'.

- The terminology 'homological convergence' is slightly misleading
  because the paper studies persistent homology, as far as I understand.

- The introduction of topological concepts in the background section
  jumps between the 'geometrical' and the 'abstract'; I understand that
  both perspectives are valid, but I would suggest to choose only
  a single one.

- 'complex clique' --> 'clique complex'

- The discussion of chains and boundaries requires more explanations (or
  could be partially relegated to the supplementary materials). For
  instance, what is as 'signed combination' in this context? If
  I understand it correctly, the paper uses the usual $\mathbb{Z}_2$
  coefficients, so there are no signs?

- I would suggest not to use $\mathbb{B}$ and $\mathbb{Z}$ to denote the
  boundary and cycle subgroups, respectively, as the later one can be
  easily confused with the set of integers or certain characteristic-$p$
  fields. The more common terminology seems to be to just use $B$ and
  $Z$ here.

- There is a distinction between the 'homology' and the 'homology
  group'. The way the paper defines it, I assume that 'homology group'
  is meant in l. 87.

- The definition of homology group needs some refinement. The sentence
  'is the quotient space of $\mathbb{B}$ a subspace of $\mathbb{Z}$' is
  confusing to me. Please rewrite this more precisely.

- The introduction of filtrations is not motivated; the 'nested family
  of simplicial complexes' could be introduced by building more
  intuition here.

- The related work discussion should, as mentioned above, delineate this
  paper from Rieck et al. [26].

- The heading 'Algebraic Topology object' is rather confusing to me;
  I would suggest to rewrite it as 'directed flag complex', for
  instance.

- As outlined above, the distance calculation lacks information about
  stability and other guarantees.

- In the experimental section, the use of the MLP should be explained in
  more detail. What does it do and how does it work exactly?

- I don't understand the 'input order' experiment; does it refer to
  ordering of samples (i.e. on the batch level)?

- The use of additional topological descriptors makes the use of the
  word 'distance' somewhat imprecise: the bottleneck and Wasserstein
  distances are metrics in the mathematical sense; while there *is*
  a well-defined metric between, for instance, persistence landscapes or
  silhouettes, this is only (to a certain extent) an approximation to
  the bottleneck/Wasserstein metrics. This distinction should be made
  clearer in the paper.

- Figure 6 should add standard deviations of the curves. An additional
  clarification of the content of this figure would also be appreciated
  by readers; I found the discussion to be slightly confusing because
  I did not get the relevance of the cumulative distance. The statement
  'the evolution of the homological convergence [...] seems to be very
  similar to the one of the validation score' also needs some
  clarification. Except for Figure 6b, where I observe some oscillation
  behaviour and maybe the tendency to converge, I don't observe
  convergence anywhere else.

- The same applies to Figures 7–9. Adding some more details here will
  result in a clearer description of the method. Maybe some of the
  curves could also be relegated to the supplementary materials?

- I would also suggest to investigate the use of energy distances or
  energy correlations, as these measures are more flexible and capable
  of assessing more than just linear dependencies between two
  variables. See [*Energy statistics: A class of statistics based on distances*](https://www.sciencedirect.com/science/article/abs/pii/S0378375813000633)
  for more details.

## Terminology and style

For the most part, the paper is written well. I have some suggestions
concerning the style.

- Please use `sort` as a `natbib` option in order to ensure that
  citations are sorted; this makes the manuscript easier to read.

- 'contained simplicial complex' --> 'nested simplicial complexes'

- can't --> cannot

- The paper employs non-standard terminology in certain places. The
  diagram, for instance, is called 'persistence diagram', no
  'persistence homology diagram'. Please ensure consistency with
  existing papers here. The same applies to capitalisation; I personally
  see no need to write 'Persistence Diagram', but if this spelling is
  used, it should be used consistently.

- 'non-cumulative homology': I think the word 'distances' is missing
  here.

- The use of 'Means mean' and 'Deviations mean' is slightly confusing;
  I would suggest to show follow the format $\mu \pm \sigma$, with $\mu$
  being the mean and $\sigma$ being some measure of variance, such as
  the standard deviation.

- For the references list, I would suggest to carefully check which
  version of a paper is being cited. Paper #24, for instance, has
  already been published as an ICML paper.

**Time Spent Reviewing:**

3

---

> ### Author Response · Authors · 2021-08-10
> **Answer to Reviewer Kand**
>
> Many thanks for the extensive review. We appreciate your insightful comments and feedback.
>
> Answering your points:
>
> - Please check the supplementary material for an extensive background. We were considerably limited with the page limit.
> -  The direction of the graph edges is important to us. We model both positive and negative weights using flag complexes.
> - Yes, we are using flag complex (directed simplicial complex). Line 166: "Algebraic Topology object:  For each weighted directed graph associated with the state of a neural network, we link a directed flag complex to it"
> - Please check supplementary material for the subsequently presented paper on distances and representations
> - Their stability with respect to perturbations on PDs has been the subject of different studies
> [6, 9].  Also studied in a recent publication:
> https://openreview.net/pdf?id=X1bxKJo5_qL
> - We will revise this section according to the 'roadmap' comment. The word you have chosen is the correct one, we are working with larger models and architectures.
> - "Rieck et al. paper analyses consecutive layers, the global topology of the network is not taken into account.
> Also, absolute values for edge weights are used. In our case, edge direction is important (for that reason we use flag complexes)
> Rieck representation cannot be used for comparing different NNs.
> - We will include a specific paragraph talking about the difference in NN representation. Please review Rieck et al. section 3.1 and compare it with our representation.
> - We have not yet defined a numerical criterion for such tasks (e.g. early stopping). This is an experimental paper where we intend to analyze the learning process. Looking at the graphs it seems intuitive at what point to stop the learning of the NN.
> - We think so, but we do not yet have a theoretical development to support the definition of such a measure. However, we think that the large topology-validation correlation is yet a very positive finding as it can enable new work to be done.
>
> Regarding the detailed comments, we will apply your suggestions, but with the following remarks:
> - Homological convergence is a way of naming the convergence in terms of persistent homology that occurs, during the learning process, in the distance between the persistent diagrams of consecutive states of the NN.
> - No, geometric and abstract homology are different things. A geometric homology considers that the points are embedded in R^n and distance comes from the space embedding the points. In this case, Vietoris-Rips is usually used to calculate the persistent homology.  On the other hand, in our case, we use an abstract homology that does not require a space embedding the points of the graph. We set up a filter that selects the axes of the graph and use Z_2 for the homology calculation (i.e. simply whether the nodes are connected or not).
> - We use flag complexes, i.e. directed simplicial complexes. The sign of the axes is important (in Rieck's paper it is not).
> In the context of line 77, "signed combination" means that the boundary operation (\partial) uses alternating signs. The formula for the boundary operation is included in the supplementary material.
> - We will include a specific paragraph talking about the difference in NN representation with respect Rieck et al. representation.
> - Stability with respect to perturbations on PDs has been the subject of different studies [6, 9].  Also studied in a recent publication:
> https://openreview.net/pdf?id=X1bxKJo5_qL
> In general, we have made 5 executions for each run in order to obtain more stable results.
> - The input order experiment is based on reordering the samples (train batch), recall that for all the runs, NNs are initialized randomly.
> - Regarding the potential imprecision of the word distance: Yes, we call one thing the distance between diagrams, and the other we call the vectorization of the diagram (an approximation of the diagram in order to calculate the distance).
> - We will add the min max average of the curves in the supplementary material.
> - We will clarify the "homological convergence" concept.
> - Please see additional material, most of the curves get stable when the validation curve stabilizes.
> - Yes. We can reorder the figure, add information and move some content to the additional material.
> - Regarding your suggestion on energy distances, thank you. It would be interesting to compare the accumulated distances with each other using these energy distances.
>
>
>
> Many thanks for all the terminology and style suggestions. We will apply all of them in the revised version.
>
> Finally, regarding the limitations and societal impact, the article focuses on MLPs. Our method is also applicable to CNNs, since they have an MLP equivalent, and also to Transformers. We are planning to further investigate the applicability of our method to these other architectures. However, we acknowledge that the proposed simplicial representation model does not seem to be directly applicable to RNNs given the existence of direct multiplications of the inputs (LTSM and GRU model). Finally, we also note that we experimented with a static CNN layer previous to the MLP.

---

> > ### Comment · Reviewer_Kand · 2021-08-11
> > **Some clarifications**
> >
> > Thanks for your response. I want to provide additional clarifications from my side:
> >
> > > Please check the supplementary material for an extensive background. We were considerably limited with the page limit.
> >
> > I fully understand. Nevertheless, my points concerning clarity and exposition may require significant restructuring. The main text of the paper should be as self-contained as possible. At present, the paper requires referring to supplementary materials and other papers to be understood. I would suggest to make this more palatable to a general ML audience.
> >
> > > The direction of the graph edges is important to us. We model both positive and negative weights using flag complexes.
> > > [...] Yes, we are using flag complex (directed simplicial complex). Line 166: "Algebraic Topology object: For each weighted directed graph associated with the state of a neural network, we link a directed flag complex to it"
> >
> > This needs to be clarified and defined. Please define a directed flag complex (my default, flag complexes are undirected), following (for instance)  [Luetgehetmann et al.](https://arxiv.org/abs/1906.10458).
> >
> > > Please check supplementary material for the subsequently presented paper on distances and representations
> >
> > Please see my comment above—all relevant materials should at least be briefly mentioned/discussed in the main text.
> >
> > > Their stability with respect to perturbations on PDs has been the subject of different studies [6, 9]. Also studied in a recent publication: https://openreview.net/pdf?id=X1bxKJo5_qL
> >
> > Please add a brief discussion on this.
> >
> > > "Rieck et al. paper analyses consecutive layers, the global topology of the network is not taken into account. Also, absolute values for edge weights are used. In our case, edge direction is important (for that reason we use flag complexes) Rieck representation cannot be used for comparing different NNs.
> >
> > I agree with everything but the last statement; the paper mentions the comparison of different NNs and, for this reason, even introduces normalisations of the proposed measure.
> >
> > > We have not yet defined a numerical criterion for such tasks (e.g. early stopping). This is an experimental paper where we intend to analyze the learning process. Looking at the graphs it seems intuitive at what point to stop the learning of the NN.
> >
> > An early stopping criterion would have the benefit of giving a direct quantification of the utility of the proposed method. Even if some readers are 'on the fence' when it comes to TDA in general, a criterion for preventing overfitting would be understood immediately. The correlation experiments are a good step, I agree, but I would suggest additional post-hoc experiments.

---

> > > ### Author Response · Authors · 2021-08-13
> > > **Answer to Reviewer Kand (II)**
> > >
> > > > I fully understand. Nevertheless, my points concerning clarity and exposition may require significant restructuring. The main text of the paper should be as self-contained as possible. At present, the paper requires referring to supplementary materials and other papers to be understood. I would suggest to make this more palatable to a general ML audience.
> > >
> > > Thanks for the suggestion and we do agree. We will try to add definitions and devote more subsections to make some points more explicit and clear in the revised version, once we have the additional page. If the reviewers point out specific points that find less clear, we will include more information about them.
> > >
> > > > This needs to be clarified and defined. Please define a directed flag complex (my default, flag complexes are undirected), following (for instance) Luetgehetmann et al..
> > >
> > > Indeed, it is a directed flag complex and in fact, we are using Flagser to compute it using the Giotto-TDA library (https://giotto-ai.github.io/gtda-docs/0.5.1/modules/generated/homology/gtda.homology.FlagserPersistence.html#gtda.homology.FlagserPersistence)
> > >
> > > > Please see my comment above—all relevant materials should at least be briefly mentioned/discussed in the main text.
> > >
> > > We agree and we will do so once we have the additional page for the revised version.
> > >
> > > > Please add a brief discussion on this.
> > >
> > > Thanks for the suggestion. We agree that it is an important point and we will have more space to discuss it once we have the additional page for the camera-ready version.
> > >
> > > > I agree with everything but the last statement; the paper mentions the comparison of different NNs and, for this reason, even introduces normalisations of the proposed measure.
> > >
> > > We acknowledge that the article worked on a similar topic, but we believe that the differences in the method are not simple technical differences, but fundamental differences that make our method "truly topological". Apart from the differences you mention (layer-wise vs globally, which is a big difference to us, and whether absolute edge weights are used): 1 - It is debatable whether topology is the right tool to analyze the connectivity of pairs of layers. 2 - We study homology groups until H_{3} and Rieck et al. only study connectivity, that is, H_{0}. 3 - Rieck et al. only see from which weight the network is connected, 4 - Rieck et al. arguably don't do a topological analysis per se, because their method is equivalent to use a minimum-spanning tree and treated as such.
> > >
> > > Quoting from Rieck et. al: "As the filtration contains at most 1-simplices (edges), we capture zero-dimensional topological information, i.e. how connected components are created and merged during the filtration. These information are structurally equivalent to calculating a maximum spanning tree".
> > >
> > > > An early stopping criterion would have the benefit of giving a direct quantification of the utility of the proposed method. Even if some readers are 'on the fence' when it comes to TDA in general, a criterion for preventing overfitting would be understood immediately. The correlation experiments are a good step, I agree, but I would suggest additional post-hoc experiments.
> > >
> > > We consider the validation curve/loss to be our baseline, and therefore we compare our method to it. We see early stopping as just yet another application of the method, not the baseline of the method itself. However, with the information and plots we provided, it is possible to see where would have stopped the training with early stopping using either validation loss or our distances. We will make it explicit for the revised version. Also, for the revised version we can try to add the differences in e.g. accuracy if using validation or our method in early stopping to better quantify the differences, as you suggested.

---

> > > > ### Comment · Reviewer_Kand · 2021-08-13
> > > > **Further clarification**
> > > >
> > > > Thanks for the replies. I want to clarify my point about existing work:
> > > >
> > > > > We acknowledge that the article worked on a similar topic, but we believe that the differences in the method are not simple technical differences, but fundamental differences that make our method "truly topological". Apart from the differences you mention (layer-wise vs globally, which is a big difference to us, and whether absolute edge weights are used): 1 - It is debatable whether topology is the right tool to analyze the connectivity of pairs of layers. 2 - We study homology groups until H_{3} and Rieck et al. only study connectivity, that is, H_{0}. 3 - Rieck et al. only see from which weight the network is connected, 4 - Rieck et al. arguably don't do a topological analysis per se, because their method is equivalent to use a minimum-spanning tree and treated as such.
> > > >
> > > > These are important claims, which should be discussed in the paper. All these claims necessitate a comparison with Rieck et al.; in my opinion, it is important to delineate how a given method improves on existing work. I fully agree with you here that this can be the case—I am not questioning the overall utility of the proposed method. If this were a *sui generis* method, there would be no need for additional comparisons. But in this case, existing work is available and addresses similar questions, so it needs to be dealt with fairly.
> > > >
> > > > I see this paper as a natural extension of the line of the work by (among others), Bianchini & Scarselli, Guss & Salakhutdinov, Ramamurthy et al., and Rieck et al. All of these papers addressed certain generalisability or complexity questions of neural networks by topological means. The closest in spirit of all of these, having experiments on generalisability etc., still appears to be Rieck et al. I think the 'burden of proof' is on the authors here to show that they improve on existing work.
> > > >
> > > > > We consider the validation curve/loss to be our baseline, and therefore we compare our method to it. We see early stopping as just yet another application of the method, not the baseline of the method itself. However, with the information and plots we provided, it is possible to see where would have stopped the training with early stopping using either validation loss or our distances. We will make it explicit for the revised version. Also, for the revised version we can try to add the differences in e.g. accuracy if using validation or our method in early stopping to better quantify the differences, as you suggested.
> > > >
> > > > I disagree with this assessment. Correlation scores are calculated based on the validation accuracy; the loss curves are depicted as-is. A baseline would have to be employed in the experiments as well, for instance by measuring to what extent the validation accuracy correlates with the validation loss (albeit this would only measure to what extent evaluation metrics and loss are aligned). To expand on this point: the paper is currently proposing a hypothesis, viz. 'Topological summaries can be useful in assessing whether a neural network generalises to the validation data set.' At present, the paper provides some hints towards this hypothesis, but the hypothesis is not subjected to a critical assessment or comparison. By this, I mean the question 'How would generalisation performance typically be assessed?' One potential answer for ML practitioners is to study loss curves. Now, one particular baseline experiment could be to say 'How well is the loss on the validation data set aligned with performance on a held-out data set?,' for instance. In this case, loss curves could be used as a comparison partner, and then the paper could quantify the performance of the new proposed method over these curves. The current experiments do not underscore this point.
> > > >
> > > > In some sense, the experimental setup is such that there will be always a topology-based method that performs well. While this is good from the point of making TDA available to a larger audience, it could also be construed as 'moving the goalposts,' since ML has been dealing with generalisation assessment *without* TDA.
> > > >
> > > > To summarise: I understand that there are differences of the proposed method in comparison to existing work. I think it's absolutely critical to provide an improved assessment of the claims purported in the paper before I can endorse it for publication.

---

> > > > > ### Author Response · Authors · 2021-08-23
> > > > > **Answer to Reviewer Kand (III)**
> > > > >
> > > > > > These are important claims, which should be discussed in the paper. All these claims necessitate a comparison with Rieck et al.; in my opinion, it is important to delineate how a given method improves on existing work. I fully agree with you here that this can be the case—I am not questioning the overall utility of the proposed method. If this were a *sui generis* method, there would be no need for additional comparisons. But in this case, existing work is available and addresses similar questions, so it needs to be dealt with fairly..
> > > > >
> > > > > > I see this paper as a natural extension of the line of the work by (among others), Bianchini & Scarselli, Guss & Salakhutdinov, Ramamurthy et al., and Rieck et al. All of these papers addressed certain generalisability or complexity questions of neural networks by topological means. The closest in spirit of all of these, having experiments on generalisability etc., still appears to be Rieck et al. I think the 'burden of proof' is on the authors here to show that they improve on existing work."
> > > > >
> > > > > Two comments:
> > > > >
> > > > > 1 - We agree that the paper needs to add certain/explicit comparisons with previous works (not necessarily quantitative), and more specifically with Rieck et al, and we will do so in the revised version.
> > > > > 2 - However, we insist that it is not clear whether Rieck et al. would constitute an actual baseline/reference for what we propose:
> > > > > a) In the selected datasets, gains by early stopping are not very clear, many hyperparameter combinations give similar accuracies.
> > > > > b) See results in Figure 4b in Rieck et al. To us, those do not mean that the early stopping has been done properly and that it is not clear whether it's actually useful in this specific case.
> > > > > c) They don't use other early stopping methods or regularizations (other potential baselines)
> > > > > d) If you take a look at the leaderboards:
> > > > > https://paperswithcode.com/sota/image-classification-on-fashion-mnist
> > > > > https://paperswithcode.com/sota/image-classification-on-mnist
> > > > > https://paperswithcode.com/sota/image-classification-on-cifar-10
> > > > > The Early Stopping technique they suggest does not even get close to SOTA.
> > > > >
> > > > > Our paper is not focused on providing an Early Stopping technique but rather to study the evolution of neural networks and show that there is a large correlation. With this in mind, different techniques could be developed. Examples: develop architectures that learn faster, select better hyperparameters, early stopping, adaptative dropout, and so on. We do not understand why we have to compare with this and to implement something that is not the core of our paper.
> > > > >
> > > > > > I disagree with this assessment. Correlation scores are calculated based on the validation accuracy; the loss curves are depicted as-is. A baseline would have to be employed in the experiments as well, for instance by measuring to what extent the validation accuracy correlates with the validation loss (albeit this would only measure to what extent evaluation metrics and loss are aligned). To expand on this point: the paper is currently proposing a hypothesis, viz. 'Topological summaries can be useful in assessing whether a neural network generalises to the validation data set.' At present, the paper provides some hints towards this hypothesis, but the hypothesis is not subjected to a critical assessment or comparison. By this, I mean the question 'How would generalisation performance typically be assessed?' One potential answer for ML practitioners is to study loss curves. Now, one particular baseline experiment could be to say 'How well is the loss on the validation data set aligned with performance on a held-out data set?,' for instance. In this case, loss curves could be used as a comparison partner, and then the paper could quantify the performance of the new proposed method over these curves. The current experiments do not underscore this point.
> > > > >
> > > > >
> > > > > We do compare the validation accuracy* (that is the standard way an ML practitioner would assess generalization) with the evolution of our metric. The validation accuracy curve correlates with our topological difference curve.
> > > > >
> > > > > *We did not use the loss as in our experiments it always keeps decreases without necessarily implying a better performance. Instead, we used accuracy.
> > > > >
> > > > > > In some sense, the experimental setup is such that there will be always a topology-based method that performs well. While this is good from the point of making TDA available to a larger audience, it could also be construed as 'moving the goalposts,' since ML has been dealing with generalisation assessment *without* TDA.
> > > > >
> > > > >
> > > > > We see this point but believe that the potential of TDA for having a better understanding of deep learning is big enough to be worth the risk of potentially "tricking" the field into "moving the goalposts" as you say. The proof of time will tell.
> > > > >
> > > > > > To summarise: I understand that there are differences of the proposed method in comparison to existing work. I think it's absolutely critical to provide an improved assessment of the claims purported in the paper before I can endorse it for publication.
> > > > >
> > > > > Again, we agree that we need to add comparisons to the previous work you say, and we will do so in the revised version.

---

### Official Review · Reviewer_RrH9 · 2021-07-10

**Rating:** 4
**Confidence:** 4

**Summary:**

This paper proposes a method to evaluate the generalization of a treined neural network using persistent homology. They measure the distance between Persistent Diagrams corresponding to Neural Networks in the training process, and show that the accumulation correlates with the Validation Curve. They also discuss which method(difinition) is appropriate for the distance between PDs.

**Limitations And Societal Impact:**

I do not believe that the content of this paper will have any negative social impact. The ability to evaluate generarization without validation set is useful for practitioners, considering the situation where it is practically difficult to collect data. Due to the above concerns, it is still unclear whether it can be used in practice.

**Main Review:**

In training NNs, the capture of generalization is very important, so we often divide the data into training set and validation set. On the other hand, there are some problems with this method, such as the dependency on the partitioning method and the small amount of training data. From these perspectives, it would be a great advantage if generality could be evaluated without a validation set. This paper represents the NN of the learning process as a weighted graph and evaluates generality by calculating persistent homology. Evaluating the network using persistent homology is a very interesting approach because it may allow us to ignore the randomness of the order of the neuron layers. On the other hand, there are some concerns about the evaluation method of Generality:
1. Since the score of the proposed method is obtained only from the training data, we cannot evaluate whether the method can be used for data whose domain is different from the training data. I think it is used to make sure that the model does not overfit the training data, but can the detection of overfit be proven by simply checking the correlation with the validation curve? In particular, it is more difficult to understand because of the normalization. I think you need more discussion.
2. In the proposed method, each score is obtained from successive NNs in the training process. Intuitively, I think this captures the fact that NNs change less with the trainning process. Since the training curve and the variation of NN are strongly correlated, isn't the proposed method looking at the convergence of the training curve? If the training set and validation set are from the same domain and there is no model overfit, the training curve and validation curve will be correlated. You may want to make sure that your experiment is not just correlating with the training curve.
3. For practical use, the proposed score has a different range of values depending on the problem. If the value of the score changes every time, it may be difficult for users to judge the generality of the score by looking at the value.

Another comment:
For example, it would be more interesting to discuss how to measure the stability of NNs based on the variation of each batch in each epoch.

Minor comments:
- The figure appears to have a solid line and a dotted line, but this is very unclear and we suggest you make it clearer.
- I think you should write what W,w is in definition of normalized weight(3) to make it easier to read.

**Time Spent Reviewing:**

6 hours

---

> ### Author Response · Authors · 2021-08-10
> **Answer to Reviewer RrH9**
>
> Many thanks for your insightful feedback and comments.
>
> Regarding your main review:
>
> 1 - We expect data from valid split to be a subset from all the data whose domain should be the same.
> Validation data is always used to monitor the performance of the network on unseen examples (estimating the generalization error)
>
> 2 - We are trying to distinguish numerical convergence from homological convergence. There are numerous studies on the numerical convergence of NNs but they cannot explain why sometimes such convergence does not occur and, in general, it is not possible to compare this process between different NNs. On the other hand, our proposal captures the global structure of the network.
> We are not plotting or using the training curve. The curves you can see are both the topological difference curve and the validation split performance curve over the time
>
> 3 - The topological complexity of the problems and of the NNs posed varies from one problem to another. It is not possible to have a universal topological measure (i.e. there is no limit for example on the number of elements of homology groups in a given dimension). What is done in the visualization is to perform an auto normalization of the values of the distances of the persistence diagrams.
>
> Regarding your "other comment": Recall that batches are not computed at the same time (first Batch A, backprop. Batch B, backprop, Batch C, backprop, ...) . You can not guarantee that the change from Batch A in T2 is exclusive to the update from Batch A T1
> Also see line 222: "Note that homological distances are obtained at the end of each batch, while validation metrics are only computed on each epoch"
>
> Regarding your minor comments:
> - Please, read de Y1 axis and the Y2 axis: Validation score (dashed) and Distance difference (line)
> - We will do that, thanks for the suggestion.
>
> Finally, regarding the limitations and societal impact, we note that our point is not only about not having a validation set, it is also about having a more principled estimation of the generalization error, which can be relevant both theoretically and in practice if this research line is further investigated.

---

> > ### Comment · Reviewer_RrH9 · 2021-08-16
> > **Additional comments**
> >
> > Thanks for your response. Here are some additional comments.
> > I understand that the task in this paper is to estimate the generalization performance of NN. The generalization performance is the accuracy for data other than the training data. My fundamental question is that even if the normalized results are correlated with the validation curve, it may be difficult to estimate the accuracy for data other than the training data.
> >
> > Regarding "another comment", My intention is "first Batch A, backprop. "calc PH",  Batch B, backprop, "calc PH", Batch C, backprop, "calc PH"..."
> >
> > The minor comment means that it is difficult to distinguish between solid and wavy lines, so it would be better to write them more clearly.

---

> > > ### Author Response · Authors · 2021-08-22
> > > **Answer to Reviewer RrH9 (II)**
> > >
> > > > Thanks for your response. Here are some additional comments. I understand that the task in this paper is to estimate the generalization performance of NN. The generalization performance is the accuracy for data other than the training data. My fundamental question is that even if the normalized results are correlated with the validation curve, it may be difficult to estimate the accuracy for data other than the training data.
> > >
> > > Correct: The idea of the paper is to propose and evaluate a new method to understand the generalization performance of NNs, but not to quantify this generalization. If you see the topological curves (PH evolution during training), when they stabilize and the topology starts to evolve less aggressively, then the network is not further learning. This is a qualitative analysis. We don't have yet a mathematical formula for exactly determining the optimal moment to stop the training.
> > >
> > > > Regarding "another comment", My intention is "first Batch A, backprop. "calc PH", Batch B, backprop, "calc PH", Batch C, backprop, "calc PH"..."
> > >
> > > This is what we do: batch A, backprop, save, batch B, backprop, save, PH A & B, Distances, batch C, save, PH B & C, Distances, batch D, save, PH C & D, Distances. We only compute validation accuracy when a full epoch has finished (all the batches have passed through the network). We can make the algorithm more explicit (using Latex algorithm typeset) on the revised version of the paper.
> > >
> > > > The minor comment means that it is difficult to distinguish between solid and wavy lines, so it would be better to write them more clearly.
> > >
> > > We will try to make these curves more clear. Thank you.

---

### Official Review · Reviewer_MNbS · 2021-07-12

**Rating:** 5
**Confidence:** 4

**Summary:**

This paper proposes a topological data analysis approach to investigate the inner workings of a deep network. In particular, the generalization performance of a deep network is measured without a holdout set by analyzing the topological change of the network during training. Multiple experiments conducted over various datasets demonstrate that the method performs reasonably for various learning rates, batch sizes, and dropout probabilities.

**Limitations And Societal Impact:**

I do not see issues related to the societal impact of this work. However, this paper is not applicable to CNNs, which are crucial for modern deep learning. This could be added to the limitations section of the paper.

**Main Review:**

PROS
The approach is really sensible. In fact, I strongly believe that the topology of training dynamics can reveal why and how a neural network generalizes. In fact, in the future, by encouraging certain topological properties during training, it may be possible discover networks that generalize better. The paper delivers a clear message and is easy to understand.

CONS
- I am not entirely convinced by the experimental evaluation given in this paper. On one hand, because of the convergence properties of the (stochastic) gradient descent algorithm,  the magnitude of the 'updates' to the weights of a neural network will decrease as training progresses. On the other hand, any successful training is expected to decrease the validation error (stochastically). Hence, it is not surprising that these quantities correlate - even if not causally related. How would this be an indicator of the generalization performance? Is there any theoretical line of thought which would justify this? From an empirical standpoint, I suggest that the authors compare different architectures. Can the PD distance identify which of these networks would generalize better?

- The proposed method seems to be agnostic to the construction of the graph. One could also construct the graph as in Corneanu [10], or potentially in many other different ways. Why do we pick the proposed method of graph construction? The impracticality claim for [10] also does not seem to be justified in this paper. Unfortunately, [2], which shares the exact similar graph construction, does not explain this either.

- This paper seems to contain extreme resemblance to [2]. If [2] (attached in supp.) is also a submission, I would doubt the contributions of this paper. I invite the area chairs to look into this issue.

- Maybe we can have the definition of Heat vectorization in the paper?

- In general it would be good to improve the captions of the figures to be more descriptive.

- Why are silhouette and heat distances the same in Table 1?

**Time Spent Reviewing:**

4

---

> ### Author Response · Authors · 2021-08-10
> **Answer to Reviewer MNbS**
>
> Thanks for your kind feedback and insightful comments.
>
> Answering the cons you have found, in the same order:
>
>
> - Yes, but it strongly correlates to validation performance (generalization capabilities). Please check on the supplementary material the unnormalized experiments regarding dropout, learning rate. This is precisely what we are studying. The topological convergence in the learning process seems to be different from the convergence of the weights in the stochastic updating of the weights.
>
> - Indeed this is what happens in the last steps of training (when NN weights vary very little implies small distances in the persistence diagrams) but the opposite is in general not true, we can have large changes in the weights of the NN that do not imply any important homological change.
>
> - There are experiments in which the generalization does not happen or the network does not learn. The generalization indicator is not currently defined mathematically because there is no clear theoretical framework. It is clear that the change of the slope of the normalized cumulative curve is related. We should say in future work that we are going to work on the definition of this indicator and its theoretical basis.
>
> - We do, MLP and CNNs with MLPs.
>
> - From what can be seen in the graphs, yes. We have found that the ones that do not have large noise or topology changes and the ones that do not have very small topology changes
>
> - We do not use the representation from Coneanu as activations are associated with input data. We are interested in studying neural structure as a function not as a function evaluated on specific data. We have questions regarding the use of activations:
> a) How much data do you pass through the network?
> b) It is not the same to pass {training, validation, testing, production} data. Isn't it?
> c) The subset of data selected may not be representative.
> d) Information flow of the network (weights) might not be taken into account completely.
>
> - [2] is focused on characterizing and comparing NNs, trained for different problems, while this paper focuses on the analysis of the learning process and the variables involved in it (dropout, learning rate, ...) for the same network applied to a specific problem. We believe we followed all rules and recommendations in the call (https://nips.cc/Conferences/2021/CallForPapers) but we are happy to discuss that with the meta-reviewer/chair if required.
>
> - It was included in the supplementary material. Otherwise, see: https://ieeexplore.ieee.org/document/7299106 or https://giotto-ai.github.io/gtda-docs/0.5.1/theory/glossary.html#heat-vectorizations"
>
> - We will fix the captions of the figures.
>
> - We strongly recommend checking the distances that are in the Supplementary Material, correlations are a way higher in some cases.
>
> Regarding the limitations and societal impact, the article focuses on MLPs. Our method is also applicable to CNNs, since they have an MLP equivalent, and also to Transformers. We are planning to further investigate the applicability of our method to these other architectures. However, we acknowledge that the proposed simplicial representation model does not seem to be directly applicable to RNNs given the existence of direct multiplications of the inputs (LTSM and GRU model). Finally, we also note that we experimented with a static CNN layer previous to the MLP.

---

### Official Review · Reviewer_YPfU · 2021-07-15

**Rating:** 4
**Confidence:** 3

**Summary:**

Summary:
The paper proposes to monitor neural networks during training by considering a momentary state of a network (MLP or CNN) as an abstract simplicial complex as defined by considering the network as a weighted directed graph. Then persistent homology features are computed resulting in persistence diagrams which are vectorized to compute distances between subsequent network states during training. The paper proposes to monitor this measure they call homological convergence to identify training stages where generalization error is low -- without the need to estimate performance by means of a validation set.

**Limitations And Societal Impact:**

Computation scalability was considered the main limitation in the paper. Otherwise, I could not find any discussion of potentially negative societal impact. For instance, when significant runtime due to scalability issues are in the focus, the paper could mention that this could affect CO2 expenditures of models employing these techniques.


**Main Review:**

Major points:
-I find it relevant and interesting to assess generalization capabilities of neural networks by means of TDA without requiring a validation set, and I am curious to see where this line of research is developing.
- The paper is clearly written, and the non-familiar reader also receives a clear introduction to TDA.
- My biggest concern is that the main finding, i.e., that PH of the network architecture allows to monitor performance without a validation set, is not novel. For instance, in Reference [26], Rieck et al employed PH of the network architecture (albeit using a weighted undirected graph) to distinguish good from poor performing networks and even proposed an early stopping scheme with a PH-based patience approach for guiding training and monitoring of neural nets without a validation set, amongst others also using MNIST and CIFAR-10. That being said, even though the empirical observation that PH of the network correlates with generalization has been known before, to my knowledge it is still an open question *why* this is the case, i.e., the theoretical workup of this phenomenon is still pending and would be quite relevant to have addressed - which I encourage the authors of this paper to investigate, as this would considerably strengthen the paper.

One way I could imagine that potential explanations of this phenomen could be identified could be via dedicated comparison studies:
e.g. do simple statistical moments of the weight distribution also correlate with generalisation, or could there be a way to identify relevant cliques in the network which are topologically relevant and also coinciding with a "lottery winning" subnetwork, etc?

- The experiments could be strenghened by the addition of certain monitoring baselines to showcase the gained value of employing PH here.
    For instance, in [26] the PH-based measure was compared to patience with validation loss (what is the difference in performance and epochs after when early stopping was initiated). Also, it would be interesting to add simple baselines measures to check that they are not already helpful alone (like non-topological information about the weight distribution).


Minor points:
- Figure 6: it took me some time to understand the legend / caption. I would suggest to repeat "# layers" in the legend, such that is easier to understand the legend without reading the caption first.
- Line 294, typo: "We posed the of question whether"

**Time Spent Reviewing:**

~ 5 hours

---

> ### Author Response · Authors · 2021-08-10
> **Answer to Reviewer YPfU**
>
> Many thanks for your insightful comments and feedback.
>
> Answering your points:
>
> - Rieck et al. paper analyses NN consecutive layers, the global topology of the network is not taken into account.
> Also, absolute values for edge weights are used. In our case, edge direction is important (for that reason we use flag complexes)
> Rieck representation can't be used for comparing different NNs.
>
> - Modestly we have to recognize that currently, we do not have the theoretical background to relate the convergence on the family of functions representing the learning of an NN (not only the weights and biases but also including the activation functions) with the convergence of the topological objects (homological persistence diagrams) that we associated with the neural network. We are very interested in the theoretical underpinning of our experimental advances.
> We are currently gathering the resources to focus on these theories in the near future but the focus of the paper was to put the first stone on this research."
>
> - It could be a good idea. In fact that "lottery winning" subnetwork could be a good idea to find the optimal representation of the NN that solves that problem. At the moment we have not gone beyond the PHs associated to the PDs (this is based on the cardinalities of these groups and not on the analysis of the elements of the group). One could try to simplify an already learned NN by making simplifications that are not topologically relevant.
> This idea could be the subject of a specific paper.
>
> - You are right in the fact that we do not have a general baseline. However, we (1) make some control experiments, (2) we apply this over many datasets, combinations of hyperparameters, and distance metrics, and (3) we do capture the topological structure of the network as seen on the subsequently introduced paper (see supplementary material) (4) there is no literature presenting a baseline.
>
> - Any matrix-based topological analysis won't take into account the relationship among neurons and thus, the structural information of the neural network won't be covered.
>
> Regarding the minor points, thanks for the suggestion and correction, we will apply/fix it.
>
> Finally, regarding the limitations and societal impact, 1) In practice, we have not observed important scalability issues. 2) We will include a discussion on CO2 expenditures. We observe that you can stop the learning motivated by both the topology and validation curves (maybe your hyperparameters are not right) and save CO2.

---

> > ### Comment · Reviewer_YPfU · 2021-08-11
> > **Discussion points**
> >
> > ```- Rieck et al. paper analyses NN consecutive layers, the global topology of the network is not taken into account. Also, absolute values for edge weights are used. In our case, edge direction is important (for that reason we use flag complexes) Rieck representation can't be used for comparing different NNs.```
> > ok, granted there are certain technical differences (whether topological features are computed layer-wise and aggregated globally - or computed globally directly, or whether absolute edge weights are used or not), however the main discovery that "PH of the network architecture allows to monitor performance without a validation set" has been shown before in Rieck et al. https://openreview.net/pdf?id=ByxkijC5FQ, which means that this contribution needs to adjusted/updated accordingly.
> >
> > ```- Modestly we have to recognize that currently, we do not have the theoretical background to relate the convergence on the family of functions representing the learning of an NN (not only the weights and biases but also including the activation functions) with the convergence of the topological objects (homological persistence diagrams) that we associated with the neural network. We are very interested in the theoretical underpinning of our experimental advances. We are currently gathering the resources to focus on these theories in the near future but the focus of the paper was to put the first stone on this research."```
> > I understand that this is challenging at this stage. I can imagine that certain basic experiments (i.e. generalization properties of a 2 layer MLP and dissecting various properties of the NN graph and weight matrices) may reveal a pattern that could be used to forward the theory as well. Maybe it even turns out that the correlation of PH with generalization is confounded by another statistical property (which may also be easier to study).
> >
> > ```- It could be a good idea. In fact that "lottery winning" subnetwork could be a good idea to find the optimal representation of the NN that solves that problem. At the moment we have not gone beyond the PHs associated to the PDs (this is based on the cardinalities of these groups and not on the analysis of the elements of the group). One could try to simplify an already learned NN by making simplifications that are not topologically relevant. This idea could be the subject of a specific paper.```
> > yes, I would look forward to it. But this is just one hypothesis, I think we need to conceive and test many more.
> >
> > ```- You are right in the fact that we do not have a general baseline. However, we (1) make some control experiments, (2) we apply this over many datasets, combinations of hyperparameters, and distance metrics, and (3) we do capture the topological structure of the network as seen on the subsequently introduced paper (see supplementary material) (4) there is no literature presenting a baseline.```
> > I am forced to disagree with this point. In my view, when a paper says: we can do X without the need for Y, then the obvious experiment would be to do X without Y in the proposed way, and compare it against the baseline / current standard: doing X the old way, using Y.
> > Since its the closest paper imo, in Rieck et al., they claim that Neural Persistence (a PH-based measure of the NN graph) can be used to assess the performance of a model without validation data and then experimentally test this by comparing Neural Persistence against the standard practice, i.e., querying a validation loss during training as for early stopping. So, my point was not "you need to compare against several baselines that were recently published", but more of a epistemological kind: if we want to show that we can do X without needing Y, then we actually compare it with Y. As a sidenote, it would have been nevertheless interesting to compare the proposed approach with Neural persistence (Rieck et al.), to check whether the mentioned differences also impact experimental results or take-aways.
> >
> > ```- Any matrix-based topological analysis won't take into account the relationship among neurons and thus, the structural information of the neural network won't be covered.```
> > I am sorry, I am not sure which of my points this is intended to answer.

---

> > > ### Author Response · Authors · 2021-08-13
> > > **Answer to Reviewer YPfU (II)**
> > >
> > > > ok, granted there are certain technical differences (whether topological features are computed layer-wise and aggregated globally - or computed globally directly, or whether absolute edge weights are used or not), however the main discovery that "PH of the network architecture allows to monitor performance without a validation set" has been shown before in Rieck et al. https://openreview.net/pdf?id=ByxkijC5FQ, which means that this contribution needs to adjusted/updated accordingly.
> > >
> > > We will nuance the contribution, but we believe that the differences in the method are not simple technical differences, but fundamental differences that make our method "truly topological". Apart from the differences you mention (layer-wise vs globally, which is a big difference to us, and whether absolute edge weights are used):
> > > 1 - It is debatable whether topology is the right tool to analyze the connectivity of pairs of layers.
> > > 2 - We study homology groups until  H_{3}  and Rieck et al. only study connectivity, that is, H_{0}.
> > > 3 - Rieck et al. only see from which weight the network is connected,
> > > 4 - Rieck et al. arguably don't do a topological analysis per se, because their method is equivalent to use a minimum-spanning tree and treated as such.
> > >
> > > Quoting from Rieck et. al: "As the filtration contains at most 1-simplices (edges), we capture zero-dimensional topological information, i.e. how connected components are created and merged during the filtration. These information are structurally equivalent to calculating a maximum spanning tree".
> > >
> > > > I understand that this is challenging at this stage. I can imagine that certain basic experiments (i.e. generalization properties of a 2 layer MLP and dissecting various properties of the NN graph and weight matrices) may reveal a pattern that could be used to forward the theory as well. Maybe it even turns out that the correlation of PH with generalization is confounded by another statistical property (which may also be easier to study).
> > >
> > > Indeed this is a huge challenge that we plan to tackle in future work.
> > >
> > > > yes, I would look forward to it. But this is just one hypothesis, I think we need to conceive and test many more.
> > >
> > > Coming back to the original point, the software we use (Giotto-TDA/Flagser) returns the cardinalities of the homology groups but we have not studied the representants of those groups.
> > >
> > > > I am forced to disagree with this point. In my view, when a paper says: we can do X without the need for Y, then the obvious experiment would be to do X without Y in the proposed way, and compare it against the baseline / current standard: doing X the old way, using Y. Since its the closest paper imo, in Rieck et al., they claim that Neural Persistence (a PH-based measure of the NN graph) can be used to assess the performance of a model without validation data and then experimentally test this by comparing Neural Persistence against the standard practice, i.e., querying a validation loss during training as for early stopping. So, my point was not "you need to compare against several baselines that were recently published", but more of a epistemological kind: if we want to show that we can do X without needing Y, then we actually compare it with Y. As a sidenote, it would have been nevertheless interesting to compare the proposed approach with Neural persistence (Rieck et al.), to check whether the mentioned differences also impact experimental results or take-aways.
> > >
> > > We consider the validation curve/loss to be our baseline, and therefore we compare our method to it. We see early stopping as just yet another application of the method, not the baseline of the method itself. However, with the information and plots we provided, it is possible to see where would have stopped the training with early stopping using either validation loss or our distances. We will make it explicit for the revised version.
> > >
> > > > I am sorry, I am not sure which of my points this is intended to answer.
> > >
> > > Our apologies, this reply was not intended for your review.

---

### Decision · Program_Chairs · 2021-09-27

**Decision:**

Reject

**Comment:**

The paper proposes a novel method for evaluating the generalization of NN using persistent homology.   While the paper introduces an interesting and novel idea, there are some weak points.  The empirical validation of the proposed idea is not strong enough. As the method is not supported by theory, more careful empirical justification is needed.  Also, there are previous works that have some overlap with this submission in the direction, such as Rieck et al. [26].  More comparison is necessary to highlight the advantage of the proposed method.
 Based on these observations, we need to judge the current paper is unfortunately not above the acceptance threshold.   However, we encourage the authors to improve the work and submit it to another conference or journal.